# Risk Assessment and Characterization in Tuna Species of the Canary Islands According to Their Metal Content

**DOI:** 10.3390/foods12071438

**Published:** 2023-03-28

**Authors:** Enrique Lozano-Bilbao, Indira Delgado-Suárez, Soraya Paz-Montelongo, Arturo Hardisson, José J. Pascual-Fernández, Carmen Rubio, Dailos González Weller, Ángel J. Gutiérrez

**Affiliations:** 1Grupo Interuniversitario de Toxicología Alimentaria y Ambiental, Facultad de Medicina, Universidad de La Laguna (ULL), Campus de Ofra, 38071 San Cristóbal de La Laguna, Spain; 2Ecología Marina Aplicada y Pesquerías, i-UNAT, Universidad de Las Palmas de Gran Canaria, Campus Universitario de Tafira, 35017 Las Palmas de Gran Canaria, Spain; 3Departamento de Biología Animal y Edafología y Geología, Unidad Departamental de Ciencias Marinas, Universidad de La Laguna, 38206 La Laguna, Santa Cruz de Tenerife, Spain; 4Departamento de Obstetricia y Ginecología, Pediatría, Medicina Preventiva y Salud Pública, Toxicología, Medicina Legal y Forense y Parasitología, Área de Toxicología, Universidad de La Laguna, 38200 La Laguna, Santa Cruz de Tenerife, Spain; 5Instituto Universitario de Investigación Social y Turismo (ISTUR), Universidad de La Laguna, 38200 San Cristóbal de La Laguna, Spain; 6Servicio Público Canario de Salud, Laboratorio Central, 38006 Santa Cruz de Tenerife, Spain

**Keywords:** metal, tuna, trophic level, ICP-OES, bioaccumulation

## Abstract

Bioaccumulation is the process by which living organisms accumulate substances, such as pesticides, heavy metals, and other pollutants, from their environment. These substances can accumulate in the organism’s tissues over time, leading to potential health risks. Bioaccumulation can occur in both aquatic and terrestrial ecosystems, and can have a significant impact on the health of both humans and wildlife. The objective of this study is to find out if the concentrations of metals in the tuna species of the Canary Islands are suitable for human consumption and if they pose a health risk. Fifteen samples of *Acanthocybium solandri, Katsuwonus pelamis, Thunnus albacares, Thunnus obesus and Thunnus thynnus* present in canaries were analyzed. Ten grams of muscle were taken from each specimen and the metals Al, Cd, Cr, Cu, Fe, Li, Ni, Pb and Zn were determined by Inductively Coupled Plasma Optical Emission Spectrometry (ICP-OES). The tuna species that presented more metals with a higher concentration compared to the others was *T. thynnus*, reaching up to 100 times more than the other studied species in Fe content with 137.8 ± 100.9 mg/Kg, which may be due to the fact that it is the largest species that reaches ages of more than fifteen years. The species *Thunnus thynnus* should not be suitable for commercialization according to the current legislation on the concentrations of Cd in blue fish, since 75% of the specimens studied exceeded the concentration legislated for Cd. A total of 40% of the studied specimens of this this species exceeded the legislated values for the concentration of Pb in oily fish meat, so this species must be monitored to ensure that it does not pose a risk to human health.

## 1. Introduction

Anthropogenic activities are the main cause of marine pollution. Residual and industrial discharges, overexploitation of fishery resources, and oil spills cause serious damage to coastlines and marine fauna. Among the different contaminants that can be found in the marine environment, metals are one of the most interesting due to their great persistence and their biomagnification through the food chain [1,2,3]. The toxic heavy metals present in the marine biota have different origins. The contribution of drainage from emerged continental and insular areas, the direct dumping of urban and industrial waste into the sea, the contribution from the atmosphere, and the underwater geological footprint itself are the main factors to be taken into account [4,5,6,7]. Biotic factors, such as the presence of organic matter in suspension, the presence of microorganisms, and the texture of sediments can also affect the biological assimilability and amplification of marine food webs. Organic matter in suspension can provide a source of nutrients for organisms, while microorganisms can break down organic matter and release nutrients into the environment. The texture of sediments can also affect the availability of nutrients and the growth of organisms [8,9,10,11]. Anthropogenic activities, such as port or industrial activities, or natural phenomena such as rock erosion, leaching or volcanic emissions, can release large amounts of metals into the marine environment. Most of the metals tend to accumulate in the bottoms, forming part of the sediments or remaining in suspension [12,13,14,15].

Multiple sources of water pollution have devastating consequences for marine life. Fish and marine mammals that are higher in the food chain are exposed to higher levels of toxins from their exposure to both contaminated water and from feeding on fish that are also exposed to them. Cadmium, mercury and lead are considered the most toxic metals for animal species and the environment [16,17,18,19,20]. These metals produce adverse biological effects in organisms, these being lethal or sub-lethal. The toxic effects of heavy metals are manifested by affecting the growth rate, physiological functions, reproduction and mortality in fish [21,22,23]. Metals can enter fish through three pathways: the gills, the digestive tract, and the skin surface. The gills are the route through which the highest rate of metal entry from the water occurs and the skin surface accounts for the lowest percentage. In the Canary Islands we have different modes of pollution: either natural, such as dust from the Sahara Desert, submarine volcanic eruptions and upwelling; or anthropogenic, such as discharges into the sea, submarine outfalls, and tourist pressure being the most important ones [24,25,26].

Tuna are found at a high trophic level in the ocean, preying on many types of large organisms such as anchovies and sardines, but also mackerel, flying fish, squid, shrimp and eels, as well as smaller tuna. The Canary archipelago is a transition zone for large tuna, so the monitoring of these species can explain the state of the ecosystem [27,28,29,30,31]. In addition, they are species of fishing interest in these islands. Environmental monitoring should also include risk monitoring, which is the determination of adverse effects on the health of consumers that may occur as a result of their exposure to food-borne hazards. The species studied in our work are pelagic fish that belong to the group of oily or fatty fish. The intake of fish from this group is generally recommended due to its content of polyunsaturated fatty acids, especially those known as omega-3 (eicosapentaenoic acid and docosahexaenoic acid) acids, whose consumption has been associated with the prevention of cardiovascular risk and the improvement of the lipid profile [32,33]. In addition, its ingestion provides proteins of high biological value, as well as a remarkable amount of vitamins (both soluble and fat-soluble) and minerals. However, consuming fish may also involve the ingestion of toxic substances that have accumulated in their tissues. There are numerous studies that show the presence of heavy metals in different species consumed by the population [34,35,36,37] in this manner. Among other toxic metal contaminants that do not fulfill any physiological function in the body, Hg, Pb and Cd can be found, among others. The legislation regulating tuna and cadmium levels are an important step in protecting consumers from the harmful effects of lead. Furthermore, it is a step towards sustainable fishing and a more responsible fishing industry. This legislation will help ensure that tuna caught are safe to eat and will help protect the oceans and their biodiversity [38]. 

To achieve this objective, the study analyzed the concentrations of metals in the tuna species of the Canary Islands, as well as the relationship between the metal content patterns of tuna by trophic level with ecological characteristics. The ecological characteristics of the tuna species will also be studied, including their feeding habits and habitat selection. Finally, the results of the study were used to determine the suitability of the metal concentrations for human consumption and the potential health risks associated with them.

## 2. Material and Methods

Seventy-five specimens of tuna found in the Canary archipelago (Figure 1) were caught by the Canarian fishing fleet in 2021 for this study. The species of the study included *Acanthocybium solandri, Katsuwonus pelamis, Thunnus albacares, Thunnus obesus,* and *Thunnus thynnus*, with fifteen specimens of each species being used in the study. The total length of each specimen was taken, and the species were identified by marine fishery biologists of the Canary Islands.

### 2.1. Sample Processing

Ten grams of muscle were taken from each specimen and homogenized. The samples were dried in an oven at a temperature of 70 °C for 24 h. They were then incinerated in a muffle oven for 48 h at 450 °C ± 25 °C until white ash was obtained. If after this time the total mineralization of the samples was not achieved (white or greyish-white ashes), 65% HNO_3_ was added to them in the fume hood, and they were subsequently evaporated on a heating plate at 70–90 °C. Once treated, they were re-incinerated in a muffle oven at 450 °C ± 25 °C until white ashes were obtained. The determination of the metal content was determined by Inductively Coupled Plasma-Optical Emission Spectrometry (ICP-OES). Table 1 shows the detection limits for each metal with the wavelengths for the measurement of each one [39].

A quality control solution was used to evaluate the accuracy of the determinations in every ten samples. The precision of the analytical procedure was evaluated by analyzing the international standard reference materials DORM-1 and DORM-5 (National Research Council of Canada). All data is presented in milligrams per kilogram, wet weight, and the metals analyzed were Al, Cd, Cr, Cu, Fe, Li, Pb and Zn. The blanks and standard reference materials were analyzed together with the samples (Table 1) [40]. As there was little sample tissue, only these metals were analyzed, leaving the analysis of Hg and isotopes for later studies.

### 2.2. Statistical Analysis

In order to verify whether there was variation in the metal and trace element content and composition, a permutational multivariate analysis of distance (PERMANOVA) was performed with Euclidean distances [41]. A unidirectional design with the fixed factor “species” with five levels of variation was used, according to of the *Acanthocybium solandri, Katsuwonus pelamis, Thunnus albacares, Thunnus obesus* and *Thunnus thynnus* species.

An analysis of principal coordinates (PCA) was used, taking as a factor the five species analyzed (*Acanthocybium solandri, Katsuwonus pelamis, Thunnus albacares, Thunnus obesus* and *Thunnus thynnus*); these being the concentrations of metals and trace elements of each sample, indicating which variable is best explained by univariate evaluations. A total of 9999 permutations underwent analysis for pairwise comparison, and thus the determination of whether or not there are significant references (*p* value < 0.05) [42]. The isotope data used were collected from published studies of the same species in the Atlantic Ocean.

### 2.3. Risk Assessment 

The risk assessment, the Acceptable Daily Intake (ADI), was determined according to the formulas given in [43]. The toxic assessment was carried out by following the recommendations of AESAN (the Spanish Agency for Food Safety and Nutrition) [44] on the consumption of fish throughout the week (three servings of 250 g) in adults with a mean weight of 70 kg and taking into account the reference values of Recommended Daily Intakes (RDIs) given by FESNAD, Estimated Daily Intake (EDIs) and Acceptable Daily Intakes (ADIs) [45] in metal (Al, Cd, Pb). For the calculations, an average weight of 70 kg was used. NOAEL is the toxicity index determined in the “toxicological evaluation” process, and F is a factor which we take as 100 since the interspecific [10] and intraspecific [10] variables are taken into account, and EDI characterizes the estimated daily intake of metal through the consumption of aquatic organism for an adult (μg Kg^−1^/day); Cmetal is the concentration of metal in organism (μg Kg^−1^) wet weight; Cons signifies the day-to-day consumption of seafood (g/day) in wet weight; and Bw is the body weight (Kg) of an adult [46].
ADI = NOAEL/F
EDI = (C metal × Cons)/Bw
MoS = IDE/IDA(1)

CDI is the chronic daily intake dose of carcinogenic elements (mg/kg/day), and carcinogenic risk (CR) is quantified by the chemical element cancer slope factor (SF). The human health risk of heavy metal intake was evaluated based on the chronic daily intake dose (CDI) for a chemical contaminant in the tuna fish over the exposure period and the fish intake quantity. CDI (mg/kg/day) was calculated using the following equation:CDI = (C × IR × EF × ED)/(BW × At)(2)

CDI is the chronic daily dose of fish intake; C is the concentration of heavy metals present in the samples (mg/kg); IR is the intake rate (104.7 g/day); EF is the frequency of exposure (three times per week = 156 days/year); ED is the duration of exposure (lifetime exposure = 30 years); BW is the body weight (kg), which we assumed to be 70 kg for an adult. AT is the mean time (AT = SD × 365 days/year). The mean daily fish consumption was set at 130 g/day, which is close to the recommended amount [47]. 

## 3. Results and Discussion

The length of *Acanthocybium solandri* was 105 ± 12 cm, that of *Katsuwonus pelamis* was 76 ± 0.13 cm, that of *Thunnus albacares* was 175 ±19 cm, that of *Thunnus obesus* was 192 ± 20 cm, and for that of *Thunnus thynnus,* which was 235 ± 41 cm, biometric data was only available for half of the specimens, since many samples were chosen by local fishermen. 

The results obtained in Appendix A are contrasted in Figure 2. This figure shows that the metal in which the species differ least significantly is Zn. These results were clearly visible in the PCA analysis, which accounted for about 95.5% of the total variability of the data (Figure 2). The ordination of the samples showed a clear difference in the heavy metal content of the *Thunnus thynnus* species with respect to the other four species. These groups appeared to be separated at a shorter distance than *T. thynnus*. Appendix A shows significant differences between the species in their metal content in most of them, which is what is reflected in the PCA with the variability with the vectors (metals and trace elements used). The metals that best explained the variability found in the data are represented as vectors in the PCA, which showed a clear increasing pattern for Zn, Cr, Cu and Fe in *T. thynnus* compared to the other species, despite the large variability between samples. However, *A. solandri* and *T. albacares* do not differ significantly in the content of Al, Cr, Li, Pb and Zn. The same occurs between *A. solandri* and *T. obesus*, although in this case, there are no significant differences in the content of Cd, Cr, Pb and Zn. The last pair of species that showed less significant differences in metal content is that of *T. albacares* and *T. obesus*, with no significant differences in Cu, Fe and Pb. 

On the other hand, those that differ most significantly are Al and Cd. In the case of Al, there is only no significant difference in the content between *A. solandri* and *T. albacares,* and in the case of Cd, there is no significant difference between *A. solandri* and *T. obesus.* It was also observed that the species that differs most significantly in metal content is *T. thynnus,* which was found to have higher values than those found in the other four species (Figure 2 and Table 2).

Table 2 shows the concentrations of the metals obtained in each tuna species, noting that *T. thynnus* presents a higher concentration of all metals compared to the other species. As regards Al, *T. thynnus* is the tuna species with the highest concentration of Al, with 19.56 ± 11.33 mg/kg (Table 2). Al concentrations in oceanic waters are low, in the order of nM, despite the fact that it is the third most abundant element in the earth’s crust and that it undergoes active transport to the oceans. The low concentrations of Al in the ocean are the result of the high reactivity of the element and its short amount of time spent in oceanic waters. Since Al bioaccumulates over the years, it is normal for the species that grows to the greatest size and age to be the one with the highest content of this metal [48,49,50,51]. One of the tissues in which Al accumulates the most is in the muscle, only being surpassed by the skin. In the present study, *A. solandri* was the species with the lowest concentration of Al, with a value of 0.515 ± 0.214 mg/kg. This may be due to the fact that, together with *K. pelamis*, it is the smallest species [52,53]. Regarding Cd, *T. thynnus* is the species with the highest concentration of Cd, with 0.051 ± 0.039 mg/kg. Rivers transport significant amounts of Cd to the oceans, from erosion processes, forming in the same large reservoirs where it is estimated that it can remain for 15,000 years. The Cd content in the oceans is around 0.1 µg/kg [54,55,56]. Cd accumulates mostly in the liver, so the high value obtained in the muscle tissue of *T. thynnus* should be monitored because the level of this metal is above the maximum content allowed (according to [57]) that sets the maximum content of different contaminants in food products, with the maximum legal limit of 0.05 mg of Cd/kg of fresh weight for fish meat, meaning that it would not be suitable for human consumption. [58] found higher concentrations of Cd in the muscle of *K. pelamis* compared to other larger tuna species, such as *T. obesus*. This has also been observed in the present study, since *K. pelamis* is the species with the second highest content of this heavy metal (Table 2). This may be due to the opportunistic diet based on invertebrates followed by *K. pelamis* [59,60], since crustaceans and mollusks are capable of accumulating higher levels of Cd than other organisms of a higher trophic level [18,61,62]. In addition, Table 2 shows the high concentrations of Zn in the different species compared to low concentrations of Cd, and this can be explained by the antagonism between these two elements, since Cd competes for the active site of Zn [63,64]. In the case of Cr, *T. thynnus* is the species with the highest Cr concentration compared to the other species, with 0.913 ± 1.263 mg/kg. Cr is one of the elements that can be found in wastewater from a wide variety of industrial processes. Its toxicity depends on the oxidation state and concentration in which it is found, and is of special importance the elimination of hexavalent chromium present in aqueous systems [65]. The concentration of Cr and its effects on teleost fish can be significantly modified by biological and abiotic variables such as water temperature and pH, the presence of other contaminating metals, and sex and tissue specificity [66,67]. *K. pelamis* has a lower concentration of this metal with 0.049 ± 0.011 mg/kg. In this case, the bioaccumulation effect is observed, since *T. thynnus* is the largest species and *K. pelamis* is the smallest. *T. thynnus* is also the species with the highest concentration of Cu, with 1526 ± 1114 mg/kg. In the sea, Cu is found at approximately 2.5 × 10^−4^ mg/L. Its presence is notably lower in the ocean the further one moves from the coast [68]. This explains why *K. pelamis* is the smallest species and is the species with the second highest concentration of Cu, as it is a species that comes closer to the coast and feeds on a greater number of crustaceans than the other species in the study. Crustaceans are rich in Cu, so *K. pelamis* can accumulate a higher concentration of this metal by ingesting them [69,70]. On the other hand, Fe is an important metal present in fish that migrate long distances, since they require strong swimming muscle tissue that requires high concentrations of blood that is rich in Fe. An Fe concentration of 137.8 ± 100.9 mg/kg has been found in *T. thynnus*, with this concentration being 100 times higher than in the other species, which can be explained by the large size that this species can grow to [71,72,73]. Regarding Pb, according to Regulation (EC) No. 1881/2006 [74,75,76], none of the species exceeded the maximum permissible concentration of 0.30 mg/kg of fresh weight for fish meat. 

The combination of contaminants and stable isotopes of carbon and nitrogen can be a powerful tool to analyze the trophic transfer of contaminants through foods [77]. This is because the isotopic composition of an animal’s tissue mirrors that of its prey, with a slight trophic enrichment of δ13C and δ15N of the order of 1‰ and 3.4‰, respectively [78,79]. δ15N is used to estimate trophic position [80], while δ13C is used to indicate relative dietary contributions from different primary sources in a food web [81]. δ13C values are typically higher (less negative) in coastal areas or benthic food webs than in pelagic food webs [81]. There are studies that have correlated stable nitrogen isotopes with concentrations of pollutants, such as Hg in fish tissues [82]. Therefore, the metal content found in the present study in the analyzed specimens is related to the values of δ13C and δ15N found in the same species in other studies (Table 3). Figure 3 shows that the species with the highest trophic level is *T. thynnus*, and this indicates that it is the species with the highest metal content due to the biomagnification process, which is the successive propagation of bioconcentration between the links in a food web [83]. The same occurs in *T. obesus*, as it has both a high metal content and a high trophic level. It is noteworthy that in the case of *T. albacares* which, despite having a high content of metals (as Table 3 shows), its trophic level is lower than that of other species of the same genus. [84] conclude that the trophic level of *T. albacares* has decreased in recent years, and that this species has high trophic plasticity related to a flexible and opportunistic diet throughout the year [30]. On the other hand, there is no relationship between the high trophic level of *A. solandri*, as can be seen in Figure 3, and the low metal concentration found in the present study. This is because *A. solandri* has a high degree of parasitism, as reported in the study by [85], and nutritional stress generated by poor diet quality or starvation can affect δ15N enrichment [86,87]. In the last species studied, *K. pelamis*, the relationship between the metal content of individuals and their trophic level was confirmed. It can be concluded that, as in other studies such as the one by [88], it has been proven that associating the transfer of trace metals with the stable isotopes of carbon and nitrogen is useful in defining the degree of the trophic transfer of pollutants.

On the other hand, the results of the metal concentrations found in the study species have been compared with the results of other studies. In the Canary Islands, this type of study on the metal concentration in marine species is important because, in addition to the contribution arising from anthropogenic contamination, the soils of the volcanic regions act as metal reservoirs [92].

Table 4 shows that the values obtained in *A. solandri* in the present study are similar to those reported in Ghana by [93], with the exception of Pb. The concentration of Pb found in Ghana exceeds the maximum content allowed (according to [94]) which sets the maximum content of various contaminants in food products, setting the maximum legal limit at 0.30 mg of Pb/kg of fresh weight for fish meat. In the case of *K. pelamis*, one can see that the concentrations obtained in the present study are the lowest, with only the Pb value not exceeding the maximum allowed in [94]. In addition, it is worth mentioning the difference between the Cd and Pb content between this study and one undertaken in the Azores [58], despite both being located in the eastern Atlantic Ocean and the islands being of volcanic origin. The study by [58] confirmed that there is an important volcanic source of Cd in the waters of the mid-Atlantic region, so commercial marine species for human consumption need to be carefully monitored. Similarly, the aforementioned study confirmed that volcanic activity accounts for an important contribution of Pb in the Azores but that, due to the low content of Pb found in the species in the study, it seems that volcanic activity does not have a measurable effect on Pb in *K. pelamis* in the Azores archipelago. It is noteworthy that the specimens of *T. albacares* in the present study have lower values in Cd and Pb than those found in Mexico and Ecuador, as can be seen in Table 5. The lowest metal concentrations are also observed in *T. obesus* in this study, compared to the results of [58,95]. The same did not occur with *T. thynnus*, where the values of the present study are higher than those found in Canada, Italy and Turkey [96,97,98]. The exceptions to this is Cd, which was surpassed by the concentration in Italy (0.051 mg/kg versus 0.26 mg/kg), and Zn, which was found in the lowest concentration in the present study. Using the tuna species studied here as bioindicators, it can generally be concluded that the Atlantic Ocean is one of the least polluted oceans in the world. 

### Risk Assessment 

Table 6 shows the consumption risk assessment for humans of the species studied for Al, Cd and Pb. To assess the toxic risk, three servings of 250 g each throughout the week have been estimated to be an adequate intake of the five species of the study. Table 6 shows the estimated daily intake of each metal for the study species (EDI), in addition to the calculation of the Margin of Safety. Metals are those with fixed ADI (Al, Cd and Pb). No metal exceeds or comes close to the MoS, so, considering the proposed intake scenario, there is no toxic risk to the health of consumers, since the reference values established by the European Food Safety Authority were not exceeded. The exception to this is Pb for *T. thynnus,* whose value is 0.85 and can present serious health risks in terms of the consumption of these species with regard to the recommended consumption guidelines. With the data obtained with regard to the EFSA, the ration of fish meat is 250 g three times a week, so none of the species except *T. thynnus* exceed the limits. For the three metals studied, the consumption of the established 750 g per week would exceed the IDE, so continued consumption throughout the life of this tuna meat can be dangerous.

Cadmium levels in muscle are below the maximum content allowed according to Regulation (CE) N^o^ 420/2011 (2011), which sets the maximum content of various contaminants in food products, establishing the maximum legal limit at 0.05 mg of Cd/kg of fresh weight for fish meat, except for *T. thynnus,* whose average concentration was 0.051 ± 0.039 mg/kg. According to these results, for the specimens of this species that exceed the maximum admissible quantities, which are 75% of the specimens studied, *T. thynnus* should be declared as a species not suitable for commercialization, since it does not comply with the regulations.

The current European legislation on cadmium in tuna is Directive 2006/66/EC, which establishes that the total cadmium in tuna must not exceed 0.3 mg/kg. This directive applies to all species of tuna, both bluefin tuna and albacore tuna. [57]. *T. thynnus* with 0.280 ± 0.188 mg/kg is close to the value 40% of the studied specimens of this species that exceeded the legislated value. Taking into account the concentrations obtained in our work, a more exhaustive study should be carried out for the species *T. thynnus,* since many of the analyzed specimens exceed the legislated value as fit for sale. The Tuna Lead Legislation is a provision of the Food and Agriculture Organization of the United Nations (FAO) that limits the use of lead in tuna weighing. This legislation was created with the goal of protecting consumers from the negative effects that lead can have on human health. This legislation is one of the main measures of the FAO to promote the sustainability of tuna fishing. For the chronic daily dose obtained, all values in all species were low, so there would be no problem, except for Al for T. thynnus, with a value of 1.4. Therefore, special surveillance and the continuous monitoring of the metal content of this species should be carried out to ensure that it does not pose a risk to the population if the AE-SAN recommendations are surpassed.

## 4. Conclusions

*Thunnus thynnus* has the highest number of metals in higher concentrations compared to the other tuna species due to the large size of the species. *T. thynnus* is the species with the highest trophic level, and this is because it has the highest metal content as a result of the biomagnification process, which is the successive propagation of bioconcentration between the links in a food web. Furthermore, this species differs most significantly with regard to the content of metals from the other species studied.

*A. solandri* was found to have the lowest concentration of the heavy metals in the present study. This may be because *A. solandri* and *K. pelamis* are the smallest of the study species. *A. solandri* and *T. albacares* do not differ significantly in the content of Al, Cr, Li, Pb, and Zn. The same is true between *A. solandri* and *T. obesus*, although in this case there are no significant differences in the content of Cd, Cr, Pb and Zn. The last pair of species that show less significant differences in metal content are *T. albacares* and *T. obesus*, without significant differences found in Cu, Fe and Pb.

Furthermore, this study shows that associating the transfer of trace metals with the stable isotopes of carbon and nitrogen is useful in defining the degree of trophic transfer of pollutants. Finally, using the species studied here (*Acanthocybium solandri*, *Katsuwonus pelamis*, *Thunnus albacares*, *Thunnus obesus* and *Thunnus thynnus*) as bioindicators and comparing metal concentrations with studies by other authors, one can conclude that the Atlantic Ocean is one of the least polluted oceans in the world.

The specimens of *Thunnus thynnus* captured in the Canary Islands present a risk to human health, since the concentrations of Cd and Pb present very high values. The EFSA recommends eating 750 g of fish meat a week, but one were to eat 750 g of *T. Thynnus* throughout their life they would have serious health problems.

The species *Thunnus thynnus* should not be suitable for commercialization according to the current legislation on the concentrations of Cd in blue fish, since 75% of the specimens studied exceeded the concentration legislated for Cd. A total of 40% of the specimens of this species exceeded the legislated values for the concentration of Pb in oily fish meat, so this species must be monitored with great effort so that it does not pose a risk to human health. Taking into account the concentrations obtained in our work, a more exhaustive study should be carried out for the species *T. thynnus,* since many of the analyzed specimens exceed recommended amounts; in addition, analyses should be carried out to verify the levels of organic pesticides and persistents that may be present in the tuna.

## Figures and Tables

**Figure 1 foods-12-01438-f001:**
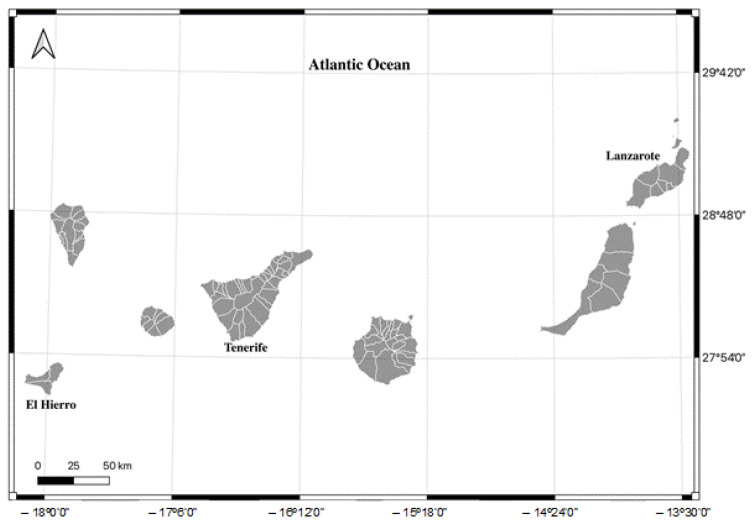
Map showing the location of the study specimens.

**Figure 2 foods-12-01438-f002:**
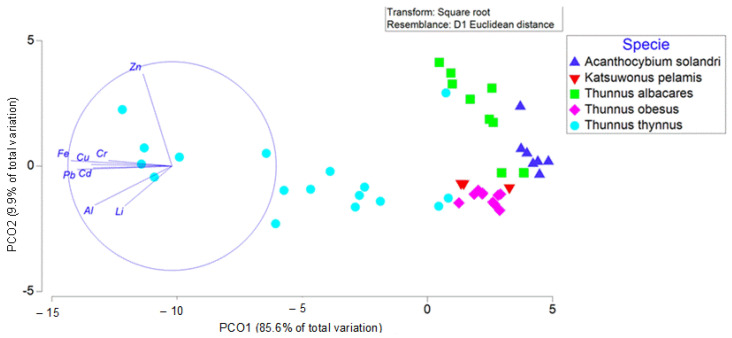
Principal component analysis (PCoA) based on Euclidean distances of square root transformed data of heavy metals and metal element content in the study species.

**Figure 3 foods-12-01438-f003:**
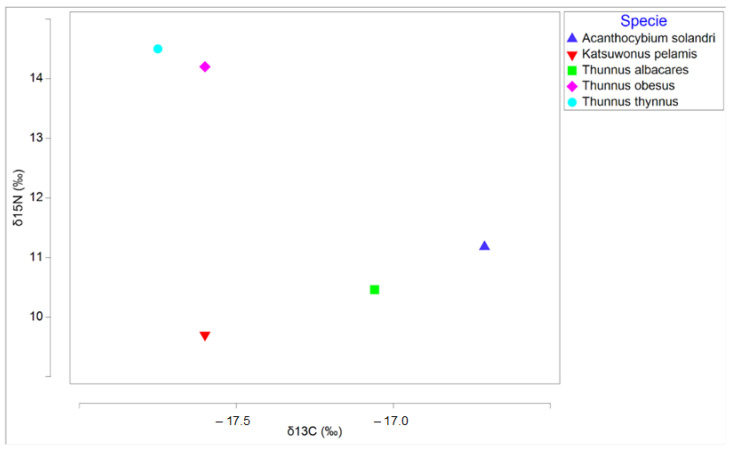
Graph showing trophic relationships taken from studies in the Atlantic.

**Table 1 foods-12-01438-t001:** Limits of detection and quantification of each metal measured by ICP-OES.

Wavelengths (nm)	Limit of D Etection (LOD) (mg/L)	Limit of Quantification (LOQ) (mg/L)
Al (167.0)	0.004	0.012
Cd (226.5)	0.0003	0.001
Cr (267.7)	0.003	0.008
Cu (327.3)	0.004	0.012
Fe (259.9)	0.003	0.009
Li (670.8)	0.005	0.013
Pb (220.3)	0.0003	0.001
Zn (206.2)	0.002	0.007

**Table 2 foods-12-01438-t002:** Metal concentrations in mg/kg for the five species analyzed (mean and standard deviation).

Species	Description	Al	Cd	Cr	Cu	Fe	Li	Pb	Zn
*Acanthocybium solandri*	Mean	0.515	0.007	0.118	0.301	4.639	0.009	0.009	9.192
	deviation	0.214	0.003	0.065	0.074	1.489	0.003	0.003	7.357
	Minimum	0.326	0.000	0.055	0.195	2.398	0.006	0.006	4.171
	Maximum	0.949	0.010	0.218	0.395	6.616	0.016	0.014	25.42
*Katsuwonus pelamis*	Mean	4.743	0.014	0.049	1.207	16.88	0.427	0.006	5.057
	deviation	0.450	0.004	0.011	0.392	7.124	0.080	0.002	0.486
	Minimum	4.161	0.010	0.034	0.702	7.705	0.324	0.003	4.430
	Maximum	5.039	0.019	0.056	1.490	21.97	0.483	0.007	5.376
*Thunnus albacares*	Mean	0.641	0.009	0.190	0.485	16.26	0.011	0.012	22.01
	deviation	0.261	0.003	0.118	0.224	8.157	0.005	0.004	17.20
	Minimum	0.283	0.006	0.065	0.282	6.330	0.006	0.005	3.920
	Maximum	0.959	0.017	0.360	0.929	29.94	0.020	0.017	50.86
*Thunnus obesus*	Mean	9.361	0.006	0.127	0.640	11.99	1.180	0.010	5.159
	deviation	2.915	0.003	0.044	0.148	3.939	0.873	0.004	0.545
	Minimum	6.006	0.004	0.095	0.453	6.761	0.211	0.005	4.434
	Maximum	13.96	0.014	0.240	0.866	19.23	3.177	0.018	6.369
*Thunnus thynnus*	Mean	19.56	0.051	0.913	1.526	137.8	1.033	0.280	14.75
	deviation	11.33	0.039	1.263	1.114	100.9	0.890	0.188	12.02
	Minimum	6.265	0.009	0.209	0.520	21.24	0.000	0.062	1.904
	Máximo	44.23	0.139	5.396	4.539	323.4	3.449	0.683	44.47

**Table 3 foods-12-01438-t003:** Values of δ13C and δ15N found in *Acanthocybium solandri, Katsuwonus pelamis, Thunnus albacares, Thunnus obesus* and *Thunnus thynnus* collected in the Atlantic Ocean. Results are expressed as mean ± standard deviation or as ranges.

Species	δ^13^C (‰)	δ^15^N (‰)	References
*Acanthocybium solandri*	−16.71 ± 0.48	11.18 ± 1.04	[85]
*Katsuwonus pelamis*	−17.6 ± 0.4	9.7 ± 1.5	[89]
*Thunnus albacares*	−17.06 ± 0.34	10.46 ± 0.34	[85]
*Thunnus obesus*	−17.6–−16.5	11.6–14.2	[90]
*Thunnus thynnus*	−17.75	14.5	[91]

**Table 4 foods-12-01438-t004:** Metal comparison of the species *K. pelamis* and *A. sonlandri* with other studies in mg/kg.

	Atlantic Ocean				Mediterranean Sea	Indian Ocean			Pacific Ocean
	Canary Islands (Spain)		Ghana	Azores (Portugal)	Spain	Iran	India	Sri Lanka	Mexico
Metal	*K pelamis*	*A. solandri*		*K. pelamis*					
Cd	0.014	0.007	0.007	0.155	0.007			0.02	0.031
Cu	1.207	0.301	0.118			4.71	2.01	4.68	
Fe	16.88	4.639					42.17	21.12	
Pb	0.006	0.009	0.054	0.152	0.03	1.1			0.055
Zn	5.057	9.192	10.01			30.57	9.23	6.5	
	The present study		[93]	[58]	[61]	[99]	[100]	[101]	[102]

**Table 5 foods-12-01438-t005:** Metal comparison of the species *T. albacares*, *T. obesus* and *T. thynnus* with other studies in mg/kg.

	Atlantic Ocean						Mediterranean Sea		Pacific Ocean		Indian Ocean
	Canary Islands (Spain)			Azores (Portugal)	Guinea	Canada	Italy	Turkey	Mexico	Equator	South Africa
Metal	*T. albacares*	*T. obesus*	*T. thynnus*	*T. obesus*		*T. thynnus*			*T. albacares*		*T. obesus*
Al	0.641	9.361	19.56			1					
Cd	0.009	0.006	0.051	0.186	0.07	0.03	0.26	0.002	0.09	0.24	0.15
Cu	0.485	0.64	1.526		2.05	1	1.15	0.819			2.13
Fe	16.26	11.99	137.8		38.7	29					42.28
Pb	0.012	0.01	0.28	0.036		0.03	0.24	0.115	0.105	0.04	
Zn	22.01	5.159	14.75		23.9	17	30.32	8.34			33.03
	The present study			[58]	[95]	[96]	[97]	[98]	[102]	[103]	[95]

**Table 6 foods-12-01438-t006:** Heavy metal risk assessment for the species studied in adults.

Metal & Limits	Index	*Katsuwonus pelamis*	*Thunnus obesus*	*Acanthocybium solandri*	*Thunnus albacares*	*Thunnus thynnus*
Al 1 mg/kg/week	EDI (mg)	0.5	1	0.05	0.5	2.09
	MoS	0.05	0.1	0.005	0.05	0.21
	kg/day	2.09	1.04	20.9	2.09	0.51
	CDI	0.34	0.67	0.03	0.046	1.4
Cd 2.5 μg/kg/week	EDI (mg)	0.0014	0.0006	0.0007	0.001	0.005
	MoS	0.07	0.03	0.03	0.004	0.27
	g/day	1.53	3.57	3.06	2.14	0.42
	CDI	0.2 × 10^−4^	0.1 × 10^−4^	0.1 × 10^−4^	0.15 × 10^−4^	0.7 × 10^−3^
Pb 0.5 μg/kg/day	EDI (mg)	0.0006	0.001	0.001	0.001	0.03
	MoS	0.01	0.03	0.02	0.03	0.85
	g/day	6.43	3.749	3.74	3.12	0.124
	CDI	0.25 × 10^−4^	0.21 × 10^−4^	0.2 × 10^−4^	0.25 × 10^−4^	0.02

**EDI**: The estimated daily intake of a food additive is the amount of a food additive ingested by the average consumer of the food; **MoS** is the difference between the quoted price of an asset and its intrinsic or real value; **ADI** is the estimation of the amount of a substance present in food or drinking water that can be consumed daily over a lifetime without an apparent risk to health., **CDI** is the chronic daily dose of fish intake.

## Data Availability

Data is contained within the article and Appendix A.

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
