# Peer review of "Risk Assessment and Characterization in Tuna Species of the Canary Islands According to Their Metal Content"

_foods, 2023, doi:10.3390/foods12071438_

Round 1

Reviewer 1 Report (Previous Reviewer 1)

Comments: 

1. Please change the title to reflect the findings or objective

2. Please included some more significant results in abstract.

3. Introduction: please mention the specific sources of pollution in the area. 

4. Method is not sufficiently described. 

5. Results: poorly presented. Add morphological features in method section. 

Statistical analyses are not relevantly used and described. PCo does not show significantly  relationship among metals. Figure 3 is hardly visible. 

For risk assessment ( as this is the main objective), show the calculation for EDI and do other test like HI, CR assessment. 

6. Conclusion:  Just use one paragraph to conclude your results which show how you have achieved your objectives. 

Author Response

  1. Please change the title to reflect the findings or objective

- Corrected in the manuscript

  1. Please included some more significant results in abstract.

- Corrected in the manuscript

  1. Introduction: please mention the specific sources of pollution in the area. 

- Corrected in the manuscript

  1. Method is not sufficiently described. 

- Corrected in the manuscript

  1. Results: poorly presented. Add morphological features in method section. 

Statistical analyses are not relevantly used and described. PCo does not show significantly  relationship among metals. Figure 3 is hardly visible. 

For risk assessment ( as this is the main objective), show the calculation for EDI and do other test like HI, CR assessment. 

  • We have added a new paragraph highlighting the PCO, and we have added the calculation formulas for Mos, ADI and EDI.
  1. Conclusion:  Just use one paragraph to conclude your results which show how you have achieved your objectives. 

- Corrected in the manuscript

Reviewer 2 Report (Previous Reviewer 2)

The sentence in the revised manuscript "Table 2 shows the concentrations of the metals obtained in each tuna species, observing that is the species that presents a higher concentration of all metals compared to the other species. As regards Al, T. thynnus is the tuna species with the highest concentration of Al, with 19.56 ± 11.33 mg/kg (Table 2). " "observing that is" maybe actually is "observing that"?

Except that, other revisions seem OK. I suggest the author check the whole manuscript to make sure there are no more language mistakes. 

Author Response

The sentence in the revised manuscript "Table 2 shows the concentrations of the metals obtained in each tuna species, observing that is the species that presents a higher concentration of all metals compared to the other species. As regards Al, T. thynnus is the tuna species with the highest concentration of Al, with 19.56 ± 11.33 mg/kg (Table 2). " "observing that is" maybe actually is "observing that"?

- Corrected in the manuscript

Except that, other revisions seem OK. I suggest the author check the whole manuscript to make sure there are no more language mistakes. 

Reviewer 3 Report (New Reviewer)

Foods

foods-2188719

Risk assessment in tuna species of the Canary Islands according to their metal content

Dear Editor,

The article deals with the investigation of the metal content of tuna species of the Canary Islands and their health risk. The topic is good. However, it needs major improvement. My specific comments and questions are appended below;

-       Please use “past tense” not “future tense” in the last paragraph of the introduction section indicating the aim of the study!

-       How were fish species identified?

-       Please give more information about the risk assessment!

-       Give the average weight of fish by species.

-       PCO or PCoA? It should be PCA!

-       Give information about C and N isotope analyses!

-       Discussion sections should be improved!

Author Response

The article deals with the investigation of the metal content of tuna species of the Canary Islands and their health risk. The topic is good. However, it needs major improvement. My specific comments and questions are appended below;

-       Please use “past tense” not “future tense” in the last paragraph of the introduction section indicating the aim of the study!

 - Corrected in the manuscript

-       How were fish species identified?

- Corrected in the manuscript

-       Please give more information about the risk assessment!

- More information has been added to the manuscript in the methodology section.

-       Give the average weight of fish by species.

- Only total length biometric data were taken, since many specimens were eviscerated. Therefore, the only real valid data we had was the total length.

-       PCO or PCoA? It should be PCA!

- Corrected in the manuscript

-       Give information about C and N isotope analyses!

- We have added the information in methodology, the data have been taken from other studies of the Atlantic Ocean, but we believe that they are of real importance for the understanding of the paper.

-       Discussion sections should be improved!

- Part of the discussion in the manuscript has been modified.

Round 2

Reviewer 1 Report (Previous Reviewer 1)

The authors have adequately addressed my comments to improve the MS.

Thank you.

Author Response

Thank you for the comments.

This manuscript is a resubmission of an earlier submission. The following is a list of the peer review reports and author responses from that submission.

Round 1

Reviewer 1 Report

The MS could have been more interesting if the results and discussion would presented clearly. 

1. Please change the title :  Metal charecterization and risk assessment of ....

2. Abstract: Mention clearly what are the risks consuming fish 

3. Introduction: lacking the gap of the study 

4. Methods: describe the study area/ sites. Pls include the morphometric information about the studied species.  Mention the accuracy  level of measurement. How did you collect the sample? Pls mention digestion procedure ( if any). 

5. Results and Discussion: we can not just start this part with table. Please re-organize your results based on objectives. 

i. Firstly mention the results and explain for each species

ii. Then you can describe statistical analyses. However, we need to explain why we are doing this analyses - PCoA..? We ca not just do unnecessary statistical with valid reason.  For risk assessment, there are some other indices you can use like HI, THQ. 

Conclusion: please show how you have achieved the objectives  in para not many paras.

Thank you

Author Response

Review

  1. Please change the title:  Metal charecterization and risk assessment of ....
  • Correct in the manuscript

  1. Abstract: Mention clearly what are the risks consuming fish 

  • Correct in the manuscript

  1. Introduction: lacking the gap of the study 

  • Correct in the manuscript
  1. Methods: describe the study area/ sites. Pls include the morphometric information about the studied species.  Mention the accuracy level of measurement. How did you collect the sample? Pls mention digestion procedure (if any). 
  • It has been added in line 157

  1. Results and Discussion: we cannot just start this part with table. Please re-organize your results based on objectives.  Done
  2. Firstly mention the results and explain for each species. Done
  3. Then you can describe statistical analyses. However, we need to explain why we are doing this analyses - PCoA.? We ca not just do unnecessary statistical with valid reason.  For risk assessment, there are some other indices you can use like HI, THQ.  Done

Conclusion: please show how you have achieved the objectives in para not many paras.

  • Correct in the manuscript
  • The results and discussion have been restructured

Reviewer 2 Report

This manuscript investigated the contamination present in tuna species from the Canary Islands, and the relationship between the patterns of metal content of tuna by trophic level with ecological characteristics. This work is meaningful, now we know we should not eat Thunnus thynnus from that area, and the Atlantic Ocean is one of the least polluted oceans in the world. The manuscript was designed well and organized well.

Author Response

Thank you very much for your review.

Round 2

Reviewer 1 Report

It can be recommended to publish